# A Systematic Umbrella Review of the Effects of Teledentistry on Costs and Oral-Health Outcomes

**DOI:** 10.3390/ijerph21040407

**Published:** 2024-03-28

**Authors:** Janneke F. M. Scheerman, Alaa H. Qari, Benoit Varenne, Harmen Bijwaard, Laura Swinckels, Nicolas Giraudeau, Berno van Meijel, Rodrigo Mariño

**Affiliations:** 1Oral Hygiene, Cluster Health, Sport and Welfare, Inholland University of Applied Sciences, 1081 LA Amsterdam, The Netherlands; 2Medical Technology Research Group, Cluster Health, Sport and Welfare, Inholland University of Applied Sciences, 2015 CE Haarlem, The Netherlands; 3Mental Health Nursing Research Group, Cluster Health, Sport and Welfare, Inholland University of Applied Sciences, 2015 CE Haarlem, The Netherlands; 4College of Dental Medicine, Umm Al-Qura University, Makkah 24381, Saudi Arabia; 5Oral Health Programme, Department of Noncommunicable Diseases, Rehabilitation and Disability (NCD), World Health Organization, 1202 Geneva, Switzerland; varenneb@who.int; 6Centre for Safety, National Institute for Public Health and the Environment (RIVM), 3720 BA Bilthoven, The Netherlands; 7Academic Centre for Dentistry Amsterdam (ACTA), University of Amsterdam, 1081 LA Amsterdam, The Netherlands; 8CEPEL, CNRS, University of Montpellier, 34090 Montpellier, France; nicolas.giraudeau@umontpellier.fr; 9Department of Psychiatry, Amsterdam University Medical Centre (UMC) and Amsterdam Public Health Research Institute, 1081 HV Amsterdam, The Netherlands; 10Parnassia Psychiatric Institute, Parnassia Academy, 2552 DH The Hague, The Netherlands; 11Center for Research in Epidemiology, Economics and Oral Public Health (CIEESPO), Faculty of Dentistry, Universidad de La Frontera, Temuco 01145, Chile; 12Melbourne Dental School, University of Melbourne, Melbourne, VIC 3052, Australia

**Keywords:** telemedicine, remote care, digital technology, teledentistry, oral health, dental, telehealth, review, digital health, effectiveness

## Abstract

Teledentistry offers possibilities for improving efficiency and quality of care and supporting cost-effective healthcare systems. This umbrella review aims to synthesize existing systematic reviews on teledentistry and provide a summary of evidence of its clinical- and cost-effectiveness. A comprehensive search strategy involving various teledentistry-related terms, across seven databases, was conducted. Articles published until 24 April 2023 were considered. Two researchers independently reviewed titles, abstracts and full-text articles. The quality of the included reviews was critically appraised with the AMSTAR-2 checklist. Out of 749 studies identified, 10 were included in this umbrella review. Two reviews focusing on oral-health outcomes revealed that, despite positive findings, there is not yet enough evidence for the long-term clinical effectiveness of teledentistry. Ten reviews reported on economic evaluations or costs, indicating that teledentistry is cost-saving. However, these conclusions were based on assumptions due to insufficient evidence on cost-effectiveness. The main limitation of our umbrella review was the critically low quality of the included reviews according to AMSTAR-2 criteria, with many of these reviews basing their conclusions on low-quality studies. This highlights the need for high-quality experimental studies (e.g., RCTs, factorial designs, stepped-wedge designs, SMARTs and MRTs) to assess teledentistry’s clinical- and cost-effectiveness.

## 1. Introduction

### 1.1. The Need for Cost-Effective Care

Due to an expanding and aging population, the global healthcare sector faces major public health challenges, such as an increase in non-communicable diseases (NCDs), a higher demand for care, healthcare workforce shortages, a lack of access to care and long waiting lists [1,2]. These challenges require a transformation of the healthcare system, as healthcare costs continue to rise, and the current healthcare system seems unsustainable in the long term [2,3]. There is a need for cost-effective innovations that increase efficiency, reduce pressure on healthcare systems and maintain the quality of care. Digital technologies offer possibilities for improving the efficiency and quality of care and supporting healthcare systems to provide cost-effective care [4,5,6].

### 1.2. Telehealth

The onset of the COVID-19 pandemic has increased the use of digital technologies, as the world transitioned to telehealth services to provide healthcare to patients remotely, while staying safe [4]. Telemedicine is a branch of telehealth, referring to the provision of healthcare services through the use of information and communication technologies in situations where a health provider and a patient (or another health provider) are not in the same location [7]. Examples of telemedicine are continuous home monitoring, involving the transfer of patient health data, and teleconsultations for remote practitioner–patient consultations.

In a recent systematic umbrella review on telemedicine, Eze and co-authors found that telemedicine interventions were at least as effective as face-to-face care [8]. Nevertheless, the authors concluded that caution is warranted, since in certain disease and specialty areas, telemedicine may be a less effective way to deliver care [8].

### 1.3. Definition of Teledentistry and Related Terms

Teledentistry is a subfield of telehealth, along with telemedicine, that is specifically dedicated to oral healthcare. The term teledentistry is often incorrectly used interchangeably for the umbrella term telehealth, which encompasses both remote healthcare delivery and the delivery of distance learning and training of healthcare providers [7]. Published dental articles have reported multiple definitions of teledentistry and related terms, and used several definitions interchangeably, like m-health, teledentistry, tele-oral medicine and telehealth.

To define teledentistry and its related definitions, the first author organized a meeting with the e-oral health network of the International Association of Dental Research (IADR) and other experts in this field [7]. According to the terminology consensus report, teledentistry was defined as follows:


*“Teledentistry represents the uses of information and telecommunication technology to provide oral healthcare services between an oral healthcare provider and a patient/recipient or other health care providers, who are separated by distance.”*


The prefix “tele” to common dental clinical disciplines describes the application of teledentistry to dental specialties, e.g., “teleperiodontology”, “teleorthodontics” and “telepaedodontics” [9]. Like telemedicine, teledentistry encompasses subunits such as tele-triage, tele-consultation, tele-assistance, tele-diagnosis and tele-monitoring [7].

### 1.4. The Current Study

The growing use of teledentistry has led to many studies investigating teledentistry applications in different settings. With the increasing number of literature reviews on teledentistry available, the next step in providing policy makers, academics and healthcare providers with evidence is to conduct a review of the existing systematic reviews, i.e., an umbrella review [10]. An umbrella review also helps to identify areas in which research is currently lacking, and to formulate recommendations for practice and for future research [10]. Consequently, the aim of this umbrella review is to synthesize existing systematic reviews on teledentistry and provide a summary of the evidence on its clinical and cost-effectiveness.

## 2. Materials and Methods

This umbrella review is consistent with the Preferred Reporting Items for Systematic Reviews and Meta Analyses (PRISMA) guidelines [11] and the umbrella review methodology described by Arosmatis and colleagues [12]. In accordance with the PRISMA guidelines, the protocol for this systematic umbrella review was registered in the International Prospective Register of Systematic Reviews (PROSPERO) on 17 May 2023 (PROSPERO 2023 CRD42023363204) with no deviation from it.

### 2.1. Data Sources and Search Strategy

Seven databases were searched from inception to 24 April 2023: (1) PubMed/MEDLINE (National Library of Medicine); (2) EMBASE; (3) Cumulative Index of Nursing and Allied Health Literature (CINAHL); (4) ISI/Web of Science; (5) Scopus, (6) Cochrane Database of Systematic Reviews (CDSR) and (7) National Health Service Economic Evaluations Database & Health Technology Assessment (NHSEED & HTA). Other sources included the major repositories of systematic reviews, including the JBI Database of Systematic Reviews and Implementation Reports, the Database of Abstracts of Reviews of Effects and the PROSPERO register. A comprehensive search strategy was designed in collaboration with a librarian. Various search terms (including synonyms and closely related words) for teledentistry/telemedicine and oral health/dentistry terms were chosen and used as Medical Subject Headings (MeSH) or free-text words in PubMed. Consequently, the search strategy was adapted and optimized for all consulted databases (See Appendix A for final search strategies). In addition, manual cross-referencing of the bibliographies of all included studies was carried out. Also, this review utilized indexing sources to retrieve subsequent relevant articles that cited the included publications. EndNote reference manager was used to store retrieved references. The search was not limited to the English language in order to maximize its sensitivity, and to identify the number of publications in other languages.

### 2.2. Eligibility Criteria

Reviews that measured the costs and oral-health outcomes of teledentistry, compared to traditional alternatives or in addition to usual care, both interventional or observational studies, were included.

#### 2.2.1. Design

We included only peer-reviewed published systematic reviews of primary studies that included a description of search terms and conducted their searches in Medline or PubMed and at least one other international scientific database. Furthermore, position papers, editorial letters, opinion pieces and books chapters were excluded due to the high risk of bias in these publications.

#### 2.2.2. Teledentistry Interventions

Irrespective of the terms used in the reviews, we included reviews in which the original studies measured the effect of teledentistry interventions that met the teledentistry definition published by the e-oral health network (see Introduction [7]). Reviews that focused on the delivery of training to oral health professionals were excluded (as this is not teledentistry, but part of telehealth). A review was included if it consisted of teledentistry interventions involving oral healthcare delivery through telecommunication systems that included either synchronous (real-time) or asynchronous (independent of time) communication between a patient (or caregiver) and their healthcare provider, or between healthcare providers, who are separated by distance.

Therefore, reviews focusing on interventions with no direct interaction with a healthcare provider, or on interventions with unidirectional communication, were excluded. For example, reviews that focused on automated text-message reminders to improve adherence were excluded (e.g., Lima et al. [13]). Also, we excluded reviews if it was unclear whether the studies included contained interventions with unidirectional or directional communication, or if the effects of these unidirectional or directional interventions were not reported separately (e.g., Al-Moghrabi et al. (2022) [14] and Fernandez et al. (2021) [15]). If a review included studies that measured the effect of oral-health or e-health interventions, besides teledentistry interventions, but the results of the effectiveness on teledentistry interventions were not reported separately, then the review was excluded [16,17].

#### 2.2.3. Population and Setting

The review consisted of studies carried out on humans, i.e., patients, healthcare providers and caregivers of any age or condition. In this review, healthcare providers (such as dentists, dental hygienists, nutritionists or nurses) are those who deliver oral healthcare. Reviews were included if the original studies utilized teledentistry within dental practice in general, specialist dental settings, a healthcare setting (such as acute care, primary or community healthcare) or home care.

#### 2.2.4. Outcomes

Reviews were included if they provided adequate information on oral-health outcomes and costs. Outputs included measures of oral health, oral hygiene levels, periodontal status, oral-health-related behaviours and oral-health-related quality of life. We excluded the psychosocial factors of oral-health behaviour (such as attitude), as these outcomes are less closely related to (better) oral health. Reviews reporting on multiple health outcomes, but separately reporting results on oral health, were included. Additionally, outcomes regarding costs included measures of economic evaluations, direct costs and indirect costs (e.g., reduced waiting lists, hours per patient, number of referrals, mean waiting times, inappropriate referrals and travel time). These outcomes were selected to reflect one of the main goals of teledentistry, which is to increase access to oral-health services while minimizing the costs and improving the quality of care. If an economic evaluation did not compare the costs and outcomes of two or more alternatives, it falls under the category of partial economic evaluation. Partial economic evaluations encompass studies focusing on (1) cost description, (2) cost-consequence description and (3) cost analysis [18]. Reviews of partial economic evaluations were also selected.

#### 2.2.5. Language

Manuscripts swritten in English, Dutch, Spanish or German were included, as the first author and one of the co-authors could understand scientific articles written in those languages.

### 2.3. Study Selection

The study selection was performed in two stages after removing the duplicates and inserting the studies into Rayyan software [19]. Deduplication was performed semi-automatically, using the deduplication capabilities in Endnote [20] and a manual check for the author and title by a medical-information specialist. In the first stage, two authors (JS and AQ) independently read the titles and abstracts of potentially relevant articles against the eligibility criteria. Disagreements were discussed and all were easily resolved through discussion. In the second stage, the full-text of the selected articles was obtained and the same two persons independently applied the eligibility criteria to confirm the final selection. Citations were coded in Rayyan as ‘included’, ‘excluded’ or ‘maybe’, as appropriate. All disagreements (indicated as ‘conflicts’ (*n* = 7) and ‘maybe’ (*n* = 37)) were resolved through discussion. If necessary (*n* = 10), a third reviewer (RM) was consulted to reach consensus.

### 2.4. Data Extraction

The first author extracted the data from the full-text articles. Other reviewing authors (AQ, BV, LS, BvM and RM) verified the extracted data. Information was extracted from each included study on the following: (1) citation details (authors’ names and year of publication); (2) objectives of the included reviews; (3) type of review; (4) participant details; (5) setting and context; (6) number of databases sourced and searched; (7) date range of database searching; (8) publication date range of studies included in the review that informed each outcome of interest; (9) number of studies, types of studies and country of origin of studies included in each review; (10) instrument used to appraise the primary studies and the rating of their quality; (11) outcomes reported that were relevant to the umbrella-review question; (12) method of synthesis/analysis employed to synthesize the evidence. The extracted data are presented in the tables and text.

### 2.5. Quality Assessment of the Selected Studies

The first author and co-authors assessed the methodological quality (critical ap-praisal) of the included studies using the validated measurement tool to assess system-atic reviews (AMSTAR 2) [21] (See Appendix A for the AMSTAR 2 checklist). According to AMSTAR 2, reviews are graded according to methodological flaws in seven critical and nine non-critical domains. AMSTAR 2 was chosen for its content validity and its ability to assess systematic reviews of both ran-domized and non-randomized studies. Before rating the included systematic reviews, each rater reviewed the AMSTAR 2 comprehensive user guide (See Appendix A for the user guide). An overall rating of each systematic review was calculated using the AMSTAR 2 online form [21]. Discrepancies among assessors were resolved via discussion until reaching consensus.

### 2.6. Data Analysis

According to the methodological approach for umbrella reviews that was reported by Aromataris et al. (2015), the analytical unit is the literature review, not the included primary studies (except when an outcome is only informed by one included study) [12]. However, we conducted thorough investigations into the original references identified by the systematic reviews. The overlap of original articles was taken into account when interpreting the data. In the results, we explicitly stated which references measured the outcomes of interests and reported whether there was any overlap. The results are presented narratively and with tables and figures for illustration. No attempt was made to compare teledentistry interventions across reviews or across review populations, and a meta-analysis was precluded due to the heterogeneity of teledentistry interventions, population and outcomes.

## 3. Results

Figure 1 presents the umbrella-review selection process in a PRISMA flow diagram. After removing duplications, the combined searches yielded 880 articles. Based on the screening of titles and abstracts, 171 full-text articles were obtained and evaluated for the preset eligibility criteria. During the full-text screening process, we excluded reviews in which we were unable to judge whether the review met the eligibility criteria due to lacking key information, i.e., important details on the design, intervention or if the outcome measures of the included studies were not clear or missing [9,22,23,24,25,26,27,28,29]. One review’s conclusion about the costs was in contrast with the conclusion stated in the original article and was therefore excluded during the data-selection process [30]. In total, 10 reviews were included in this umbrella review [31,32,33,34,35,36,37,38,39,40].

Table 1 presents an overall description of the included reviews characteristics and findings. All included reviews employed qualitative analysis to synthesize the evidence. In only one study, the primary research aim aligned with our study aim and thus exclusively addressed the outcome of interest [37]. Conversely, the remaining studies covered multiple outcomes and the outcomes of interest were reported as secondary outcome measures.

The modality of the teledentistry intervention, i.e., as synchronous (real-time) or asynchronous (store-and-forward) interactions [7], was only reported in two included reviews [31,39], of which tele-consultation was a common real-time interaction. Aquilanti et al. 2020 [31] stated that the asynchronous (store-and-forward) model was always the less costly one, followed by the real-time model and face-to-face dental visits. Also, Joshi et al., 2020 [39], stated that real-time consultation had better outcomes concerning costs.

### 3.1. An Overview of the Outcomes Reported by the Teledentistry Reviews

#### 3.1.1. Oral-Health Outcomes

Two reviews assessed the effectiveness of teledentistry on oral-health outcomes compared to traditional alternatives [32,37].

The review by Ben-Omran et al. (2021) [32] reported that there was no significant difference between intervention and control groups in terms of Geriatric Oral Health Assessment Index scores, measuring the oral-health-related quality of life. Despite positive findings, Ben-Omran and his colleagues concluded that there was insufficient evidence to firmly advocate for the long-term clinical effectiveness of teledentistry.

The review by Estai et al. (2018) [37] included nine articles considering various clinical outcomes, of which three studies specifically addressed the clinical outcome of interest, i.e., DFS scores, periodontal indices and oral hygiene scores. Despite the diverse objectives, methodologies and outcome measures employed across the included studies, teledentistry interventions were comparable to, or had advantages over, non-telemedicine approaches. However, Estai and his colleagues’ overall conclusion was that there is not yet enough conclusive evidence of the effectiveness and long-term use of teledentistry.

#### 3.1.2. Economic Evaluations and Costs

Ten reviews focused on outcomes with regard to costs and economic evaluations; of these, only five [31,32,35,36,37] provide reports on costs as outcomes of interest in their reviews and included a review of studies of true economic evaluation as defined by Drummond and collaborators (2005) [18]. The review conducted by Aquilanti and collaborators [31] included three studies reporting economic evaluations. Estai and collaborators’ [37] review also included three studies reporting economic evaluations. Those two reviews used the Drummond and collaborators’ checklist [18] to evaluate the quality of the economic evaluations. Interestingly, although these two reviews were conducted in different years, they included different studies and only one overlapped. Moreover, for the overlapping study, despite employing an identical assessment tool (Drummond et al.’s), each review yielded disparate scores.

A more recent review by Emami and collaborators [36] identified seven studies which include some elements of economic evaluation and used the level of evidence according to the Oxford Centre for Evidence-Based Medicine [41]. On the other hand, Daniel and collaborators [35] and Ben-Omran and collaborators [32] included two studies each that were found to be economic evaluations, but no evaluation tool was mentioned to facilitate the quality-evaluation task.

Interestingly, overall, these ten reviews encompassed a total of five different studies that incorporated some form of economic evaluation. These evaluations included a range of economic impacts and study designs; cost descriptions; cost analysis; cost-minimization analysis; and even time-effectiveness, which is not typically considered a form of economic evaluation on its own.

The remaining four reviews [34,38,39,40] did not report on true economic evaluations, but summarized information on costs. This information mentioned costs, mostly as the reduction of costs by using teledentistry in terms of the reduction of travel and waiting times, and of other features of telehealth that are related to costs (e.g., reduced loss of productivity, less unnecessary travel, fewer accommodation expenses and reduced time for services). Furthermore, these reviews included studies on the reduction of cost, but were not related to any health outcomes, and found that teledentistry is likely to be a cost-effective or a cost-saving alternative compared to the standard practice of face-to-face consultation. Two of those reviews [38,39] mentioned that the cost-effectiveness of teledentistry was an assumption based on the reduction of cost, but without providing any evidence of studies that substantiated claims regarding cost-effectiveness. Conversely, another study [35] emphasized the need for further economic evaluation in teledentistry to support any contention on cost-effectiveness. The review conducted by Estai and collaborators [37] indicated that cost-minimization was said to be conducted in two studies; however, a closer examination of the original studies does not support that statement. Instead, the economic analyses in the original studies were more like cost-analysis studies. Furthermore, the review by Estai et al. concluded that none of the studies included considered cost–benefit, cost-effectiveness, cost–utility or incremental economic analyses.

### 3.2. Results of Quality Appraisal

According to the AMSTAR-2 quality-assessment tool, all reviews were of critically low or low quality (see Table 2). The most common weakness across the reviews was a poor description of the original studies (e.g., missing important information on outcomes, study design or intervention). According to the authors of the included reviews, the original articles were descriptive in nature and provided a poor quality of evidence, which in turn contributed to the poor reporting quality of the reviews. The absence of good-quality economic studies in teledentistry has also been cited in the included reviews (e.g., [35,37]).

For the majority of reviews, the AMSTAR-2 quality scores were low for the following domains: Q3 (reviews did not explain the selection of the study design), Q7 (reviews did not justify the reasons for exclusions or search limitations), Q8 (reviews did not describe the studies sufficiently), Q13 (reviews did not take into consideration the risk of bias in the interpretation of the results) and Q14 (reviews did not explain how heterogeneity across studies may have impacted the results). None of the included reviews reported on the sources of funding for primary studies (Q10).

## 4. Discussion

The aim of this umbrella review was to synthesize the existing systematic reviews on teledentistry and provide a summary of evidence of teledentistry’s effects on oral-health outcomes and costs.

### 4.1. Main Findings

In this umbrella review, ten articles were selected. While most reviews described positive effects of teledentistry on oral-health outcomes, it is important to interpret these findings cautiously, due to their overall critically low quality, according to AMSTAR-2 criteria. The benefits of teledentistry and piloting teledentistry have been explored widely; however, studies focusing on the clinical effectiveness of the use of teledentistry and immediate clinical implications are limited.

Teledentistry is commonly perceived to be cost-saving and cost effective; however, this is often only based on assumptions. Although some reviews explored the health economic implications related to teledentistry, most of them based their conclusions on low-quality observational studies without clear methodologies or with small sample sizes, or on self-reports or the subjective opinions of participants. These designs do not allow for evidence-based conclusions on their cost-effectiveness. Clinical trials, particularly RCTs and other quasi-experimental designs, are in general best suited for assessing the efficacy and cost-effectiveness of an intervention, and thus allow for strong evidence-based conclusions. However, digital health interventions are complex interventions and some evaluation methods, for example, RCTs, may not be the most effective way of evaluating digital technology deployments in healthcare [42]. While RCTs provide valuable data on the clinical efficacy and safety of a particular intervention, they may not capture its economic implications and long-term outcomes. When assessing cost-effectiveness, effectiveness trials may be more suitable as they provide insights into the real-world performance of the intervention.

Hrynyschyn et al. (2022) conducted a scoping review to identify alternatives to RCT’s as potentially more appropriate evaluation methods of digital health interventions [42]. According to the authors, factorial designs were mostly used to evaluate digital health interventions, followed by stepped-wedge designs, sequential multiple assignment randomised trials (SMARTs), and micro randomised trials (MRTs) [42]. Some of these methods allow for the adaptation of interventions (e.g., SMART or MRT) and the evaluation of specific components of interventions (e.g., factorial designs) [42].

Economic evaluations are essential for informing policymakers and dental practitioners about the feasibility, benefits and challenges of integrating teledentistry into healthcare systems. Researchers and healthcare professionals increasingly acknowledge the importance of assessing the cost effectiveness, return on investment, and overall economic viability of teledentistry. The growing interest in teledentistry emphasizes the need for economic evaluations to comprehend its financial implications and benefits when integrated into oral-healthcare systems [9,35]. Nonetheless, Mariño et al.’s (2013) [9] assertion remains relevant: there is still a scarcity of studies that adequately address the issue of the economic evaluation of teledentistry. Our research demonstrated a comparatively restricted number of cost-analyses and economic assessments specifically targeting teledentistry (*n* = 10) when compared to other sectors of telemedicine. To illustrate, Bergmo’s systematic review spanning 1990–2007 identified only 33 economic evaluations within telemedicine [43]. Importantly, the economic evaluations in teledentistry have shown low adherence to Drummond’s standards [18]. This observation aligns with the quality assessments found in reviews of telemedicine.

To the best of our knowledge, this study is the first ever umbrella review to focus on the effects of teledentistry on costs and oral-health outcomes. Our findings are consistent with similar reviews conducted on both the clinical outcomes and cost-effectiveness of telemedicine [8]. Eze et al. [8] reported in their umbrella review on telemedicine that 92% of the included systematic reviews were of a low to critically low quality, measured with the same quality-measurement assessment tool as used in our umbrella review (AMSTAR). They reported that telemedicine can be cost-effective and can improve clinical outcomes, such as glycaemic control in diabetic patients and improving patients’ diet quality and nutrition. However, they found that generalizability was also hindered by poor quality and reporting standards.

### 4.2. Quality Assessment of the Included Reviews

To assess the quality of the included reviews, we used the AMSTAR2 quality-assessment tool. All the included systematic reviews were of “low” or “critically low” quality, resulting in a high risk of bias for our findings. The low quality of reviews available in the literature was mostly caused by poor reporting quality or unavailable data (e.g., missing details on the outcomes, interventions or study designs of the included studies).

Selection bias may have occurred when we had to exclude a great number of reviews (*n* = 9) that lacked key information, such as when the description of the studies was too poor and data extraction was not possible to perform. This was mostly the case when the primary research aim of that study had a different focus than our study aim. The majority of the included reviews stated that most of the original articles included in the reviews were also poorly reported. To improve the quality of future research, researchers could use the guidelines for reporting research that can be found on the website of ‘Enhancing the QUAlity and Transparency Of health Research’ (EQUATOR network) to ensure that they report all key information.

### 4.3. Strengths, Limitations and Recommendations

An advantage of an umbrella review is that it identifies gaps in a specific research field and can inform future research [44]. A disadvantage of an umbrella review is that the validity of the findings depends on the quality of the eligible systematic reviews and meta-analyses. A limitation of our umbrella review was the low quality of the included reviews according the AMSTAR2 assessment, which influences the validity of the findings of this umbrella review. Although AMSTAR2 is a validated tool, some modifications could improve its validity. For example, this assessment tool does not consider the study designs of the original studies included in the systematic reviews. However, the quality of the review can be influenced by the study designs and the robustness of the results.

Studies have reported multiple definitions on e-health and used several definitions interchangeably, such as m-health, teledentistry, tele-oral medicine, and telehealth. Another strength of this review is that we used the definition of teledentistry defined by the e-oral health network of the IADR and used this terminology to label the intervention in the reviews, rather than the terms stated in the article [7]. This is important because although the authors employed a wide search in seven databases to include all articles using alternative terms for teledentistry, it is possible that we missed articles employing alternative terminology. A recommendation arising from this review is therefore the use of universal terminology to label interventions in future research, as it would prevent different uses of terminology in future studies, increasing the homogeneity of studies and preventing publication bias. The use of terminology proposed by the e-oral health network [7] could help in achieving this objective. 

Teledentistry interventions are highly diverse in both context and applications. Given the heterogeneity of primary study samples, interventions and design, and the lack of key information on the content of the teledentistry interventions and outcomes, subgroup analyses and meta-analyses were not possible, limiting the findings. Nevertheless, we have provided a narrative review by outlining the current evidence on the effectiveness of teledentistry and highlighting the research gaps for future studies. Categorizing interventions in future studies by the types of teledentistry applications [7] and their modalities (store-forward or real-time) can be useful to provide a better overview and compare the results across studies in the future.

A high prevalence of oral diseases and a high level of unmet oral health needs are common among people living with disabilities and mental health illnesses, and among institutionalized/hospitalized people and older people [45]. Moreover, poor oral-health status is a marker of adverse health outcomes and health inequalities, indicating that enhancing access to dental care is crucial in these patient groups, for example, people with poor access to care [45].

Oral-health professionals could remotely assist other healthcare professionals to provide oral-health care in people with poor access to care. These people could greatly benefit from teledentistry applications, like teleconsultation with small cameras. However, our umbrella review shows that teledentistry has rarely been applied, and no experimental study on the effectiveness have been performed on these groups of the population with special healthcare needs [40]. Thus, especially these populations, there is a need for more high-quality experimental studies measuring the effects of teledentistry on oral-health-related outcomes and costs.

Artificial intelligence (AI) has evolved tremendously in recent years, and the application of AI in teledentistry has the potential to revolutionize remote dental care [46]. Machine learning, including deep learning-based algorithms, has been developed to create predictive models of risk assessment and diagnostic services for oral health, which can enable teledentistry to better its remote screening, diagnoses, record keeping, triaging, and monitoring of dental diseases [7,46]. We, therefore, assume that in the future, teledentistry with the integration of AI can play a bigger role in improving efficiency and quality of care, and to support healthcare systems providing cost-effective care. To reach this objective, research will also need to be stepped up in this AI area so that cost-effective interventions can be implemented.

## 5. Conclusions

Ten reviews researched the promising potential of teledentistry; however, evaluations on cost and oral-health outcomes are scarce. Only five studies incorporated some form of economic evaluation. These evaluations included a range of economic impacts and study designs; cost descriptions, cost analysis, cost-minimization analysis, and even time-effectiveness, which is not typically considered a form of economic evaluation on its own. Thus, there is insufficient qualitative evidence to support conclusions on the cost-effectiveness or long-term effectiveness of teledentistry.

The main limitation of our review stems from the low or critically low quality of the included reviews, as per the AMSTAR-2 criteria. Many of these reviews drew conclusions from studies of low quality with poor design reporting and a descriptive nature, thereby contributing to the overall poor reporting quality within the included reviews.

Therefore, high-quality experimental studies (e.g., RCT’s, factorial designs, stepped-wedge designs, SMARTs and MRTs) on the clinical- and cost-effectiveness of teledentistry are needed to increase the body of evidence regarding the digitalization of oral care.

A recommendation for future research is the universal use of terminology on teledentistry and its related terms. Using the terminology created by the e-oral health network and used in this study will help to standardize approaches and compare results across studies.

## Figures and Tables

**Figure 1 ijerph-21-00407-f001:**
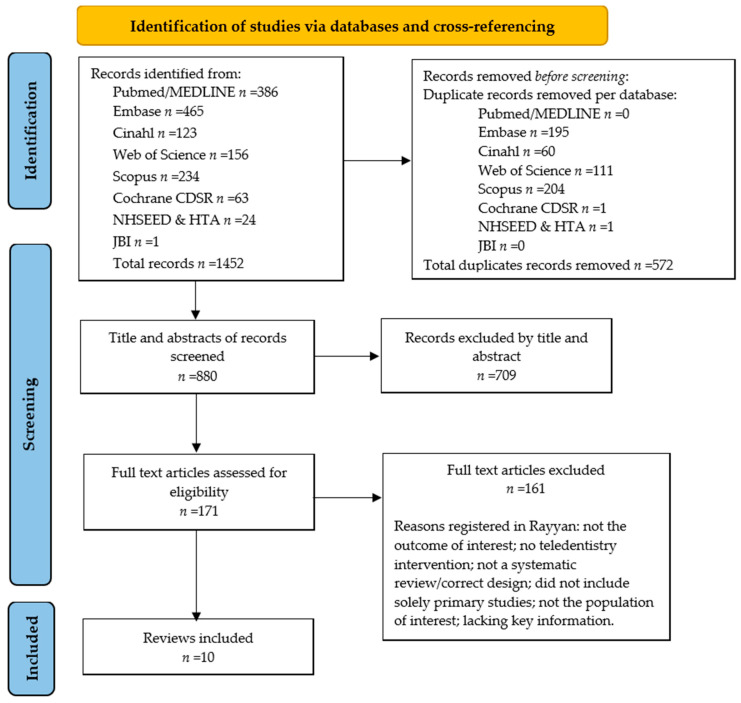
A PRISMA flow diagram of the umbrella review.

**Table 1 ijerph-21-00407-t001:** Included review characteristics and findings.

Reference	Type of Review and Study Objective Mentioned in the Review	Teledentistry Modalities and Application Researched	(4) Participant Details; (5) Setting and Context.	(6) Number of Databases Sourced and Searched; (7) Date Range of Database Searching; (8) Publication Date Range.	(9) Number of Studies, Types of Studies and Country of Origin of Studies Included in Each Review.	(10) Instrument Used to Appraise the Primary Studies and the Rating of Their Quality.	Main Findings
Aquilanti et al. (2020) [31]	The aim of the **systematic review** is to assess the feasibility of teledentistry in the provision of oral healthcare to older adults living in residential aged-care facilities. In particular, the review focused on the evaluation of the accuracy and the effectiveness of teledentistry compared to traditional face-to-face dental visits, the patient acceptability, and the costs related to the implementation of oral-health information technology provision.	Both synchronous and real-time teledentistry. Results for different types of teledentistry applications are not mentioned.	(4,5) Studies involving elderly people in nursing homes, in communities, or within in-home assistance were included. Young persons were excluded.	(6) *n* = 5; PubMed, Cochrane Library, Web of Science, Scopus and CINAHL databases. (7) Until 30 June 2020 (8) 2024–2018	(9) Six studies were included in the review, but only three studies measured the outcome of interest (two measured the effects of teledentistry on indirect costs and one measured cost analyses). Types of studies measuring the effects on costs and indirect costs included a pilot study with a cost analysis; at 6 months, a quality-improvement study and cost analysis and a cost-analysis comparison study; a multicentre, cross-sectional study; and a mixed-methods comparative study. The included studies were performed in Australia, France and Germany.	(10) The quality of the studies included in the review was evaluated by the two independent reviewers using the protocol described by Hailey et al. The overall quality score/the strength of evidence was defined by both the performance and study design. The review included mostly articles with poor or poor to fair quality, characterized by substantial limitations in the study and only one with fair to good quality. The quality assessment of the studies that included a cost analysis was performed in accordance with the Drummond et al. 10-point checklist.	The review included three studies reporting on the economic evaluations of teledentistry. Teledentistry was found to be as cost-effective as traditional face-to-face dental examinations.
Ben-Omran et al. (2021) [32]	The aim of the **scoping review** was to systematically explore and describe the literature on various uses of teledentistry in older adults, including its reported effectiveness and limitations.	Both synchronous and real-time teledentistry. Types of teledentistry applications researched are tele-consultation, tele-diagnoses and tele-intervention.	(4) Older adult population (≥60 years) (5) Medical and dental settings: academia, private practice, community clinics or hospital (nursing home, dentist practice, pharmacy, community dental clinic, hospital, academic institution, long-term facilities, primary care clinics and private clinics).	(6) *n* = 9 PubMed/MEDLINE (National Library of Medicine), Cochrane Library: Database of Systematic Reviews, Cochrane Library CENTRAL, Embase, Scopus, Web of Science Core Collection, Cumulative Index of Nursing and Allied Health Literature (CINAHL), Health Technology Assessment database, and National Health Service Economic Evaluations Database. (7) Searches were conducted in January 2020 (8) and limited to articles published from 1991 through to 2020.	(9) *n* = 19, (of which *n* = 4 measured the effects of teledentistry on costs—only one study included a cost analysis). Types of studies: non-rct; cross-sectional; rct; and observational with mixed retrospective and prospective designs. Countries: Japan, United States, Northern Ireland, China, Australia, United Kingdom, Brazil, France, India, Germany, Finland, and Portugal.	(10) The instrument used to appraise the primary studies was not mentioned in the article. The only mentioned in the discussion the overall rating of their quality (unclear how this was measured): “A limitation was the quality of the studies included, as many were cross-sectional studies with no clear methodology stated, non-RCTs with small sample sizes, or clinical trials that were dependent on self-reports or subjective opinions of participants or their caregivers.”	The authors identified cost reductions as a result of reducing avoidable dental visits to nurses with the guidance of a remote-dentist model. No significant difference was found between intervention and control groups in terms of Geriatric Oral Health Assessment Index scores, measuring the oral-health-related quality of life. Despite positive findings, Ben-Omran and his colleagues concluded that there was insufficient evidence to firmly advocate for the long-term clinical effectiveness of teledentistry.
Da Costa et al. (2019) [33]	The purpose of this **integrative review** was to collect information regarding the inclusion of the application of teledentistry tools in the public dental-health services.	Types of teledentistry applications researched are tele-diagnosis and tele-screening.	(4) a wide range of dental-patient groups, including paediatric, orthodontic and elderly patients, as well as prisoners. (5) Dental public-health services, including dental-health programs or dental-health-related actions taken at a community, state or federal level.	(6) Searches were conducted on five electronic databases (PubMed/Medline, Virtual Health Library, CINAHL, Scopus and Web of Science); (7) studies that were published from 2007 to June 2019 were included. (8) Publication date range: 2007–2018	(9) Twenty-four studies were included, of which four measured the outcome of interest: economic evaluation (two in paediatric dentistry, one in older adults and one in oral medicine). Types of studies included economic evaluations, exploratory descriptive studies, mixed-method comparative studies; cost-minimization analyses; cross-sectional studies. Country-of-origin of studies: Australia and Brazil.	(10) Due to the variety of research methods employed in the included studies, the mixed-methods appraisal tool (MMAT) was used to assess their quality. Among the 24 studies that met the eligibility criteria, 7 studies could not be assessed using MMAT because they did not have enough information regarding the methods and criteria that were employed; however, the remaining 17 studies were assessed using MMAT. Most of them (14 studies) had good-quality scores, meeting three or more of the four criteria. Furthermore, three studies were considered to have moderate-quality scores, meeting only two of the four criteria.	The authors concluded that teledentistry is cost-effective; however, no in-depth economic design is presented.
(Da Costa) Flores et al. (2020) [34]	The purpose of this **systematic review** is to summarize information on the use of teledentistry in the telediagnosis of oral lesions.	The type of teledentistry application and modulation were not mentioned.	(4, 5) Dental-clinic community patients (*n* = 41)	(6) Four databases: PubMed, Embase, LILACS (Latin American and Caribbean Literature in Health Sciences and SUMSearch. The CAPES (bancodetes.capes.gov.br/) and Google Scholar databases were used to identify additional grey literature. (7) articles published until December 2018. (8) Range: 1999–2018; the included study was performed in 2010	(9) Eleven studies were included, of which only one feasibility study performed in New Zealand reported on the outcome of interest;	(10) The bias risk and quality analyses of the study were performed independently by two authors using the Quality Assessment of Diagnostic Accuracy Studies questionnaire. The original study presented good quality, as 12 out of 14 questions were answered with yes.	The authors concluded that teledentistry is likely to be a cost-effective alternative compared with the standard practice of face-to-face consultation. However, this contention is not supported for any economic evaluation.
Daniel et al. (2013) [35]	The purpose of this **systematic review** is to identify clinical outcomes, healthcare utilization and costs associated with teledentistry.	Types of teledentistry applications researched are tele-triage and tele-screening.	(4) In the review of Daniel et al., there is no data-extraction table present nor did the text describe the participants’ details, setting and context for each original study, so we are unable to give a precise overview of the participants details of the original studies. Mentioned in the text: preschool urban children and orthodontics.	(6) Literature searches were conducted in 15 databases: PubMed/Medline, EMBASE, CINAHL with Full Text, PsychINFO, EBM Reviews (e.g., Cochrane Database of Systematic Reviews, ACP Journal Club, Database of Abstracts of Reviews of Effects, Cochrane Central Register of Controlled Trials, Cochrane Methodology Register, Health Technology Assessment and NHS Economic Evaluation Database), Scopus, Education Resource Information Center (ERIC), Google Scholar and Turning Research into Practice (TRIP). (7) Publication date from the earliest available date for each database to March 2012. (8) Publication dates ranged from 2009–2019	(9) Nineteen studies were included, of which four of the included original studies measured the outcome of interest. Cost-analysis and comparative effectiveness study. The country of origin of the cost-analysis study is United Kingdom	(10) The instrument that was used to appraise the primary studies and rate their quality was not described in the review. The discussion stated the following: Common methodological weaknesses in these studies included the lack of blinding of dentists, patients or assessors. While in teledentistry it is not always feasible to design studies with patients and dentists who are not aware of the group assignment, the use of outside assessors reduces the potential for evaluation bias. Many of the studies used convenience samples based on the geographical location of patients or patient preference, clearly introducing the possibility of selection bias. A total of 12 studies (60%) had sample sizes of fewer than 20 subjects, and only 1 of the studies provided power calculations. Small sample sizes can lead authors to conclude that no significant difference exists between groups, i.e., a type-II error, whereas the study may have insufficient power to identify a significant difference. Nevertheless, larger studies often remain challenging to carry out, as many of the teledentistry programs are still in their pilot phases and there is often a limited availability of the patient population concerned.	In terms of economic evaluation, one study concluded on the cost-effectiveness of the teledentistry approach.
Emami et al. (2022) [36]	This **systematic review** evaluated the literature on patient satisfaction with e-oral healthcare in rural and remote communities.	The teledentistry application researched is tele-consultation. Most studies used teledentistry consultations, either live or store-and-forward.	(4) Participant details not reported; (5) in rural and remote settings.	(6) Searches were carried out in four databases: Cochrane Central Register of Controlled Trials, MEDLINE, EMBASE and Global Health. (7) date range of database searching: published between 1946 and 2021; (8) publication date range: studies carried out in 1998 and 2019	(9) In total, 16 studies were included in the review, of which 7 studies focused on the outcome of interest.The types of studies comprised non-randomized clinical trials, observational studies, pilot intervention studies and cost analyses. In total, five studies were from Australia, three from India, two studies were conducted in the USA, two in Spain, one in Canada, one in the UK, one in Italy, and one in Finland.	(10) The risk of bias using the ROBINS-I risk-of-bias assessment tool for non-randomized studies. Thirteen of the selected studies were found to have a moderate risk of bias, and two other studies had critical risk in the overall assessment. One article was found to be ineligible for performing risk for bias assessment using the ROBINS-I tool. The majority of studies (11 out of 16) were considered level 4 and 3b.	Only a few studies reported the cost per unit of outcomes gained; rather, the level of satisfaction was related to reduced waiting time, the number of visits, travel, and the cost of care for patients. The review also commented on the heterogeneity and inconsistency of methodologies of the studies reviewed in terms of study design, perspective, sampling, setting, etc.
Estai et al. (2018) [37]	This **systematic review** of the benefits of teledentistry aims to inform decisionmakers who are doubtful about the capability and merit of integrating teledentistry into routine health services by presenting an objective overview of good-quality evidence for the effectiveness and economic impact of teledentistry.	Studies were clustered into two major applications, telediagnosis and teleconsultation.	(4) The majority of the reviewed studies were solely focused on the specialty of oral medicine, paediatric dentistry and orthodontics. (5) The majority of the reviewed articles did not explicitly report the setting of the study (rural or urban); however, it appears that studies were carried out in either urban or rural settings such as hospitals, clinics, childcare centres or workplaces.	(6) *n* = 3; PubMed, EMBASE and CINAHL databases (7) Through November 2016 (8) 2001–2016	(9) *n* = 6; This review included three studies that performed economic evaluations. Of these, two studies were deemed to be of fair to good quality. The review included nine articles considering various clinical outcomes, of which three studies specifically addressing on the clinical outcome of interest, i.e., DFS scores, periodontal indices and oral hygiene scores. The studies included in the review were conducted in seven different countries, with the majority of studies from Europe (*n* = 5) and the USA (*n* = 3), with one each from Japan, India and Australia	(10) The quality of each study, other than those aspects related to economic analysis, was evaluated independently by two authors using the protocol established by Hailey et al., taking into account the study performance and study design.	Despite the diverse objectives, methodologies and outcome measures employed across the included studies, teledentistry interventions were comparable to, or had advantages over, non-telemedicine approaches. However, Estai and his colleagues’ overall conclusion was that there is not yet enough conclusive evidence for the effectiveness and long-term use of teledentistry.
Irving et al. (2017) [38]	This **qualitative systematic review** aims to explore the quantitative and qualitative framework associated with the effectiveness of teledentistry in an effort to uncover the interaction of multiple influences on its delivery and sustainability.	The teledentistry application researched is tele-consultation.	(4) General dental patients/orthodontics, oral-surgery patients, hospital-referral patients and adults with tetraplegia. (5) Dental practice in both general and specialist dental settings	(6) Literature searches were conducted in nine databases: MEDLINE, Embase, CINAHL, PsychINFO, AMED, EBM Reviews, ERIC, Global Health and PREMEDLINE databases. We also searched the grey literature. (7) Database searches were conducted on 5 January 2015. (8) Publication date range: 2001–2013	(9) In total, 19 studies were included, but only 4 studies measured the outcome of interest. Study type: practitioner cohort, patient cohort and controlled trial. The country of origin of studies included in each review: UK (*n* = 2), Spain (*n* = 2) and USA.	(10) A modified Downs and Black criterion scale, which examines validity, bias, power and other study attributes, was used to assess the methodological quality of the included papers. They modified the original Downs and Black scale, as described and recommended in prior methodological systematic reviews, to exclude items that were not applicable to the designs of eligible studies. For example, items specific to randomized trials were removed for observational studies. A percentage quality score was calculated by dividing the total score received by the maximum score possible for each study. The majority of included studies were only rated as being of fair quality. The majority of the studies were reported on by the developers of the programs, creating a possible opportunity for a bias in the reporting of the results included from the studies.	The review concluded that teledentistry is a cost-saving alternative to conventional practice. However, the reduction of costs and cost-effectiveness is assumed, as no actual reviews of economic evaluation in teledentistry were provided.
Joshi et al. (2021) [39]	The aim of the **scoping review** was to identify the challenges, scope and assessment approaches of teledentistry from an Indian perspective.	Both synchronous and real-time teledentistry. Types of teledentistry applications researched are tele-consultation, tele-diagnoses and tele-screening.	(4, 5) Not described	(6) *n* = 3; Google Scholar, PubMed/Medline and Scopus; (7) searched from April to August 2020; (8) publication dates ranged	(9) Twenty studies were included in the scoping review. Only five studies reported on the outcome of interest. Types of studies were not reported; however, the review did report on the type of analyses. Analyses that have been carried out included cost-minimization analyses, cost-effectiveness analyses, model-based and cost-effectiveness analyses, and teledental asynchronous patient assessments and remote real-time oral examination. The review did not report on the countries of origin of studies.	(10) It did not assess the rigor or quality of studies. Note: outcomes were not described in detail, E.g., it was stated “Teledentistry is a cost-saving”, but no details on the design or outcome were reported.	The authors concluded that the use of teledentistry is potentially cost-effective and cost-saving compared to traditional dentistry. However, none of the studies conducted in India provide any support for that assumption.
Uhrin et al. (2023) [40]	The aim of the **systematic review** was to collect available data on how oral medicine could benefit from teledentistry solutions, and to investigate whether teledentistry could provide a reliable diagnostic method compared with clinical oral examination (COE) in the diagnosis of oral potentially malignant disorders.	Virtual examination.	(4, 5) The review included adults with suspected oral lesions. One of the included articles that measured the outcome of interest included patients of a special care clinic, with intellectual disability, cerebral palsy, Down’s syndrome, autism, seizures, HIV, liver disease, neurologic disorders, stroke or schizophrenia; the other article included <18-year-old patients referred to the clinic with oral lesions. The mean age of the population was 47 (*n* = 29) and 50 (*n* = 33).	(6) Three databases (Medline, EMBASE and CENTRAL); (7) date of searching: until November 2021. (8) Publication dates ranged	(9) Thirteen studies were included; however, only two studies investigated the outcome of interest: time effectiveness. These studies included an observational study and a cross-sectional study. The review included a meta-analysis for the primary outcome, but not for the secondary outcome. These were only described narratively. One study was conducted in the US and the other one in Brazil.	(10) Risk of bias was assessed using the QUADAS-2 tool. Certainty of evidence was evaluated based on the Grades of Recommendation, Assessment, Development and Evaluation (GRADE) workgroup’s recommendations. Four articles were excluded due to a lack of data. The QUADAS-2 tool showed that most of the domains had a low risk of bias.	The authors performed a meta-analysis on the primary outcomes; however, no statistical analysis could be performed on the secondary outcome’s time-effectiveness. One of the original studies measured the difference in time during in-person examinations (mean: 4.2 min, SD: 1.6) and virtual examinations (2.83 min, SD: 1.0).

Note: The data provided in this table are based on incomplete information gathered from the included systematic reviews, which in turn restricts the completeness of the conclusions drawn and limits the ability to conduct a comparative analysis.

**Table 2 ijerph-21-00407-t002:** Methodological quality of included systematic reviews.

Review First Author (Year)	Q1	Q2	Q3	Q4	Q5	Q6	Q7	Q8	Q9	Q10	Q11	Q12	Q13	Q14	Q15	Q16	Overall Quality
Aquilanti et al. (2020) [31]	Y	P	N	P	Y/U	Y	N	N	N	N	N/A	N/A	N	N	N/A	Y	Critically Low Quality
Ben-Omran et al. (2021) [32]	Y	N	N	N	Y	N	N	N	N	N	N/A	N/A	N	N	N/A	N	Critically Low Quality
Da Costa et al. (2019) [33]	N	N	N	P	Y	Y	N	N	U/N	N	N/A	N/A	N	N	N/A	Y	Critically Low Quality
(Da Costa) Flores et al. (2020) [34]	N	P	N	P	Y	Y	N	N	U	N	N/A	N/A	N	N	N/A	Y	Critically Low Quality
Daniel et al. (2013) [35]	Y	N	N	N	N	N	N	N	N	N	N/A	N/A	N	Y	N/A	N	Critically Low Quality
Emami et al. (2022) [36]	Y	Y	N	P	Y	Y	N	Y	Y	N	N/A	N/A	Y?	N	N/A	Y	Low Quality
Estai et al. (2018) [37]	Y	N	Y	P	Y	Y	N	Y	Y	N	N/A	N/A	Y	Y	N/A	Y	Critically Low Quality
Irving et al. (2017) [38]	Y	N	N	P	Y	Y	N	Y	Y	N	N/A	N/A	Y	Y	N/A	Y	Critically Low Quality
Joshi et al. (2021) [39]	N	N	N	N	Y	Y	N	N	N	N	N/A	N/A	N	N	N/A	N	Critically Low Quality
Uhrin et al. (2023) [40]	Y	P	N	P	Y	Y	N	Y	Y	N	N/A	N/A	Y	Y	N/A	Y	Low Quality
**Assessment Questions**																	
Critical domains (*grey highlight*)	Q2: Protocol registration Q4: Adequacy of the literature search Q7: Justification for excluding studies Q9: Risk of bias from studies included in review Q11: Appropriateness of meta-analytical methods Q13: Consideration of risk of bias in the interpretation of results Q15: Assessment of presence and likely impact of publication bias.																
Non-critical domains (*no highlight*)	Q1: Inclusion of PICO elements in review question Q3: Explain selection of study design Q5: Duplicate study selection Q6: Duplicate data extraction Q8: Description of studies Q10: Report sources of funding for primary studies Q12: Impact of risk of bias assessment on evidence Q14: Explanation for heterogeneity Q16: Report potential conflicts of interest and funding sources by review authors.																
**Grading criteria**																	
Y (Yes): Criterion met; “P” (Partial yes): Criterion partly met; “N” (No): Criterion not met; “N/A” (Not applicable), U (Unclear)																	
High Quality	No or one non-critical weakness																
Moderate Quality	More than one non-critical weakness																
Low Quality	One critical flaw with or without non-critical weaknesses																
Critically Low Quality	More than one critical flaw with or without non-critical weaknesses																

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
