# Peer review of "A Systematic Umbrella Review of the Effects of Teledentistry on Costs and Oral-Health Outcomes"

_ijerph, 2024, doi:10.3390/ijerph21040407_

Round 1

Reviewer 1 Report

Comments and Suggestions for Authors

Teledentistry refers to the use of information and communication technologies, including the transmission of clinical information and images between an oral health professional and a patient or between two oral health professionals. The purpose of this manuscript is to provide evaluations of the impact of teledentistry, quality of care, and costs. In the parts covered, the text is clear, relevant to the sector, well structured and with coherence between the parts: abstract, methods and conclusions. Overall, there are no methodological inaccuracies highlighted, also showing sufficient in-depth analysis; no self-citations are found. The conclusions generated by this overview of the literature, despite the poor quality of many studies, can be used to improve clinical practice, contributing in a complete and logical manner to implementing knowledge on the potential effectiveness of teledentistry which materializes in the diagnostic field and in follow-up; as the lack of high-quality information on the effectiveness of this technique has often been reported as a major obstacle to its implementation.

•The data reported, although referring to incomplete original studies, are consistent with the discussion. A positive note is the continuous reminder of caution when using the results indicated.

• Note table figure 1, the exclusion criteria of the relevant articles could be added.

• Weak point, the reference to reviews of studies conducted before the Covid-19 pandemic, as the landscape of research and healthcare practices is constantly evolving, the studies could lead to new information on tele-dentistry.

Reviewer 2 Report

Comments and Suggestions for Authors

The absence of a comprehensive explanation regarding the cost-effectiveness of teledentistry and its impact on oral health outcomes is noted. It is imperative to underscore that while teledentistry may streamline consultations, it should not be perceived as a substitute for substantive dental treatment. The efficacy of teledentistry in this regard necessitates further exploration.

During stage 1 (title-abstract selection), conflict resolution methodologies employed were not elucidated. Furthermore, specific articles utilized in the primary synthesis were not clearly delineated, leading to ambiguity regarding potential overlaps.

The results section lacks a detailed comparative analysis of the identified articles, and the metrics used to assess effectiveness remain unspecified.

The discussion section is convoluted and lacks clarity. Additionally, it fails to address why cost-effectiveness cannot be adequately assessed through standard clinical trials, nor does it expound on established methodologies for such evaluations.

A noteworthy limitation is the influence of article type on ROB, particularly regarding diagnostic metanalysis, which may yield less robust results compared to randomized controlled trials (RCTs).

The assessment of risk of bias should precede the discussion of results, while strengths and limitations should be addressed towards the conclusion of the discussion.

The conclusion section is overly lengthy and should succinctly summarize the study's findings and implications in a few sentences.

Comments on the Quality of English Language

English language and style are fine/minor spell check required.
